# Involvement of Mitochondria in the Selective Response to Microsecond Pulsed Electric Fields on Healthy and Cancer Stem Cells in the Brain

**DOI:** 10.3390/ijms25042233

**Published:** 2024-02-13

**Authors:** Arianna Casciati, Anna Rita Taddei, Elena Rampazzo, Luca Persano, Giampietro Viola, Alice Cani, Silvia Bresolin, Vincenzo Cesi, Francesca Antonelli, Mariateresa Mancuso, Caterina Merla, Mirella Tanori

**Affiliations:** 1Division of Health Protection Technologies, Italian National Agency for Energy New Technologies and Sustainable Economic Development (ENEA), Via Anguillarese 301, 00123 Rome, Italy; arianna.casciati@enea.it (A.C.); vincenzo.cesi@enea.it (V.C.); francesca.antonelli@enea.it (F.A.); mariateresa.mancuso@enea.it (M.M.); 2Great Equipment Center-Section of Electron Microscopy, University of Tuscia, Largo dell’Università snc, 01100 Viterbo, Italy; artaddei@unitus.it; 3Department of Women’s and Children’s Health (SDB), University of Padova, Via Giustiniani 3, 35128 Padova, Italy; elena.rampazzo@unipd.it (E.R.); luca.persano@unipd.it (L.P.); giampietro.viola.1@unipd.it (G.V.); alice.cani@unipd.it (A.C.); silvia.bresolin@unipd.it (S.B.); 4Pediatric Research Institute (IRP), Corso Stati Uniti 4, 35127 Padova, Italy

**Keywords:** cancer stem cells (CSCs), glioma, medulloblastoma, normal human astrocyte, CD133 protein, mitochondria dysfunction, transcriptomics, filopodium-like protrusions

## Abstract

In the last few years, pulsed electric fields have emerged as promising clinical tools for tumor treatments. This study highlights the distinct impact of a specific pulsed electric field protocol, PEF-5 (0.3 MV/m, 40 μs, 5 pulses), on astrocytes (NHA) and medulloblastoma (D283) and glioblastoma (U87 NS) cancer stem-like cells (CSCs). We pursued this goal by performing ultrastructural analyses corroborated by molecular/omics approaches to understand the vulnerability or resistance mechanisms triggered by PEF-5 exposure in the different cell types. Electron microscopic analyses showed that, independently of exposed cells, the main targets of PEF-5 were the cell membrane and the cytoskeleton, causing membrane filopodium-like protrusion disappearance on the cell surface, here observed for the first time, accompanied by rapid cell swelling. PEF-5 induced different modifications in cell mitochondria. A complete mitochondrial dysfunction was demonstrated in D283, while a mild or negligible perturbation was observed in mitochondria of U87 NS cells and NHAs, respectively, not sufficient to impair their cell functions. Altogether, these results suggest the possibility of using PEF-based technology as a novel strategy to target selectively mitochondria of brain CSCs, preserving healthy cells.

## 1. Introduction

In recent years, pulsed electric fields (PEFs) have become a promising clinical treatment in oncology. The effectiveness of PEF-based technologies such as electrochemotherapy, gene electrotransfer, and calcium electroporation in treating tumors depends on their ability to facilitate the uptake of chemotherapeutic agents, therapeutic genes, and calcium into the tumor tissue [1,2,3,4,5]. In contrast, other PEF treatments, such as irreversible electroporation, high-frequency irreversible electroporation, and nanopulse stimulation, are primarily based on the ablative effect of the PEF itself leading to cell death [6,7,8,9,10]. Some preview studies have demonstrated that the application of PEFs can lead to a susceptible effect in a cell-specific manner. The selectivity of the treatment was based on cell-specific lethal electric field thresholds, attributed to the different morphology of the cells [11,12,13].

To potentiate the offer of electric-mediated therapies and their effects, our group started to study the action of PEF on brain cancer stem cells (CSCs) representing the main cause of such tumor recurrence and relapse against which current approaches are ineffective [14,15,16]. In this context, our previous studies demonstrated that a specific electric pulse protocol, PEF-5 (0.3 MV/m, 40 μs, 5 pulses), showed a different impact on medulloblastoma (MB) D283 CSCs [4] compared with glioblastoma (GBM) CSCs (U87 NS, grown as neurospheres) [15]. In D283 cells, PEF-5 exposure triggered high levels of cell death associated with irreversible electroporation and a process of radiosensitization, resulting in complete eradication of tumor masses in vivo [14]. Conversely, PEF-5 exposure induced reversible electroporation in U87 NS, with consequent low levels of cell death and induction of neuronal differentiation, resulting in radioresistance but decreasing the clonogenic capacity and the ability of cells to migrate [15]. Normal human astrocytes (NHAs), used as representatives of healthy cells and present in the surrounding area of brain tumors, maintained their viability after undergoing reversible electroporation induced by PEF-5 [14]. The three cell models selected for this study exhibit different CD133-positive cell content. The CSC phenotypes of both MB and GBM were demonstrated in previous characterizations conducted by our group [4,15].

The present study aims to elucidate the PEF-5 selective mechanism in the different cell types, previously mentioned, as it results in a vulnerable or resistant response. To achieve this goal, an ultrastructural analysis was performed to fully understand specific cellular responses to PEF. Considering the plasmatic membrane as a main target of PEFs [2,14], scanning and transmission electron microscopy (SEM and TEM, respectively) analyses were performed to study changes on the cell surface, pointing our attention to transmembrane protein CD133. We investigated the morphological alterations in the cytoskeleton and internal compartments, including organelles such as mitochondria, after exposure. Specifically, the actin cytoskeleton is organized into various structures, including stress fibers, lamellipodia, filopodia, microvilli, retraction fibers, and stereocilia [5]. Among these structures, filopodium-like protrusions have a role in several cellular processes, including adhesion, intercellular communication, and motility [5,17]. In tumor cells, they are believed to contribute to highly invasive behavior and therapy resistance [18,19]. Finally, a transcriptional analysis corroborated SEM and TEM outcomes and provided us with a comprehensive understanding of the specific susceptibility or resistance mechanisms that were activated in response to PEF-5 exposure across the various cell types. Our findings suggest that mitochondrial organelles are the elective target of PEF-5, supported also by intracellular ATP evaluation and mitochondria membrane depolarization results.

It is worth mentioning that mitochondria play a crucial role in the maintenance of cell survival and CSC resistance [20]. Some studies suggest that CD133 expression may be influenced by environmental conditions and stress responses, such as hypoxia and mitochondrial dysfunction [21,22]. CD133 is a cell surface marker expressed in CSCs and is widely used as a representative marker for human stem and progenitor cells [23,24,25,26,27]. CD133-positive CSCs exhibit characteristics such as unlimited self-renewal capacity and the ability to drive tumor initiation and progression [26,28]. Additionally, high CD133 expression is closely associated with resistance to chemotherapy and radiation therapy, as well as increased survival [28,29,30]. This study highlights that the impact of PEF-5 on mitochondria is in correlation with CD133-positive cell content observed in the different cell types. The altered mitochondrial functionality, induced by PEF-5, might explain the selective action of electric exposure on brain CSCs and NHAs. Moreover, mitochondrial dysfunction could be employed as a strategy to induce the death of CSCs. In fact, despite CSCs representing a small fraction of tumor tissues, they are frequently associated with aggressive, heterogeneous, and therapy-resistant tumors [31,32,33,34,35] that still represent a difficult oncologic challenge.

## 2. Results

### 2.1. SEM Analysis: PEF-5 Exposure Induces Filopodia Disappearance and Cell Swelling

SEM analysis was performed 1 h after PEF-5 exposure to evaluate specific effects on the cellular surface. The analysis highlighted a round cell shape in all sham-exposed samples, with membrane filopodium-like protrusions distributed on the whole cell surface (Figure 1a,c,e). In each cell line, they present distinct morphologies: in NHAs and D283 cells, they are filiform, while in U87 NS cells, they are shorter and tightly bundled.

One hour after PEF-5 exposure, the filopodium-like protrusions disappeared in all analyzed cells (Figure 1b,d,f), leaving a smooth cellular surface, particularly evident in U87 NS cells, where non-uniformly distributed pores were clearly observed (Figure 1f, inset). This phenomenon was associated with a rapid and severe swelling observed in NHAs and U87 NS cells (30% and 44% respectively; Figure 1b,f,g) that was milder in pulsed D283 cells (11%; Figure 1d,g). 

### 2.2. PEF-5 Exposure Alters CD133 Protein Expression and Its Localization in MB CSCs

Given that the plasma membrane is the primary target of PEFs, we conducted an analysis on the transmembrane protein CD133. To sustain the hypothesis that CD133 protein can be a molecular mediator of the differential response of PEF-5, we analyzed CD133-positive cell proportion by flow cytometry. This protein was detected in 15% of NHA, 90% [4] of D283, and 29% of U87 NS cells (Appendix A). The flow cytometric analysis showed a significant decrease of CD133-positive cell proportion at 1 h and 24 h after PEF-5 exposure in D283 cells (Figure 2a–c) and no significant perturbation was observed in U87 NS cells (Figure 2d–f) compared with relative controls. Immunogold electron microscopy (IEM) analysis was performed to localize the CD133 protein at the ultrastructural level in D283 cells. Gold particles were detected on the plasma membrane of the cellular surface protrusions only in sham-exposed cells (Figure 2g,h). When pulsed cells were analyzed, no filopodia were observed and only rare gold particles along the flat regions of the plasma membrane were noted 1 h after treatment (Figure 2i,j).

### 2.3. TEM Analysis: PEF-5 Induces Further Cytoskeleton Alterations

Ultrastructural analysis of sham-exposed cells showed rounded shapes and confirmed the presence of filopodium-like protrusions on the surface of all analyzed cells. The cytoskeleton was uniformly distributed in the cytoplasm and cytoplasmic organelles, such as mitochondria, the endoplasmic reticulum (ER), and the Golgi apparatus, and vesicles appeared to have a healthy physiological morphology (Figure 3a,d,g). One hour after PEF exposure, TEM images confirmed the absence of membrane filopodium-like protrusions in both NHAs and CSCs (Figure 3b,e,h). Moreover, TEM analysis evidenced alterations in cytoskeleton distribution within all PEF-5-exposed cells (Figure 3b,c,e,f,h,i); a rearrangement of the cytoskeleton with a well-organized network of densely packed intermediate filaments (10 nm in diameter) was often observed close to the nucleus region (highlighted as the colored area in Figure 3b,c,e,f,h,i). In addition, 24 h after PEF-5 exposure, fluorescent F-actin staining highlighted a normal distribution of actin filaments in NHAs (Appendix A). On the contrary, in D283 and U87 NS cells, a persistent retraction of filopodium-like protrusions was associated with a disorganized distribution of actin inside the cells (Appendix A, respectively). 

### 2.4. TEM Analysis: PEF-5 Exposure Activates MVB Formation in NHAs and Affects Mitochondria in MB and GBM CSCs

In NHAs, 1 h after pulse exposure, TEM analysis highlighted a significant increase in vesiculation (14-fold PEF-5 vs. sham, Appendix A), with lamellar bodies inside the larger vesicles and microvesicles. Multivesicular bodies were also found on the cell surface, involved in a process of extrusion from the cell, suggesting a significant increase in multivesicular body (MVB) formation (Appendix A). In contrast, a small alteration in vesiculation was observed in D283 cells (PEF-5 vs. sham 1.4-fold; Appendix A) and a negligible alteration in U87 NS cells (Appendix A). Interestingly, in both NHA and U87 NS cells, the ER appeared to be significantly enlarged (PEF-5 vs. sham 2.6-fold and 2.5-fold, respectively; Appendix A), and only a negligible alteration was observed in D283 cells (PEF-5 vs. sham 1.04-fold; Appendix A). This effect appears to be proportional to the different swelling of these cells. Morphological analysis of TEM images of pulsed NHA and U87 NS cells revealed that 34% and 36% of cells, respectively, had at least a damaged mitochondrion (Figure 4a,b,e,f). All pulsed D283 cells (100%) presented damaged mitochondria, with stroma and cristae nearly entirely disorganized (Figure 4c,d). In addition, analysis at the single-cell level revealed 100% damaged mitochondria in each analyzed D283 pulsed cell (Figure 4g); damaged mitochondria were only sporadically observed in each analyzed NHA and U87 NS pulsed cell (3.6% and 2.4%, respectively, Figure 4g). Further, the mitochondrial area in U87 NS cells was significantly smaller, reduced by 30% compared with that of the control group. (Figure 4h).

### 2.5. Mitochondrial Function Assessment

TEM ultrastructural images of PEF-5-exposed cells highlighted altered mitochondrial morphology in the analyzed cells, which was particularly consistent in D283 cells, and milder in NHAs and in U87 NS cells, suggesting a possible mitochondrial dysfunction as an answer to PEF-5 exposure. To support this hypothesis, we evaluated the level of intracellular ATP 30 min after PEF-5 exposure. The luminescence fold change with respect to relative control revealed a significant depletion of ATP in pulsed D283 cells, with a 90% intracellular reduction of ATP (Figure 5a). Furthermore, PEF-5 exposure did not induce any significant decrease of intracellular ATP in NHAs, and a decrease was observed in U87 NS cells but it was not statistically significant. In addition, cytofluorimetric analysis using JC-1 as a specific sensor of mitochondrial potential was performed to evaluate if there was a correspondence between mitochondrial membrane depolarization and the level of ATP depletion. The introduction of JC-1 dye highlighted a mitochondrial membrane potential disruption after PEF-5 exposure in both D283 (Figure 5b,c) and U87 NS cells (Figure 5e,f). In fact, results showed a significant decrease in the red/green fluorescence intensity ratio that was particularly strong in D283 cells (reduced 13-fold with respect to relative control (Figure 5d)), and reduced 1.5-fold in U87 NS cells (Figure 5g), confirming a total dysfunction of mitochondria in MB CSCs as a prompt reaction after PEF treatment.

### 2.6. Transcriptional Modulation Induced by PEF-5 Exposure in NHAs, MB, and GBM CSCs

To evaluate eventual changes in the activation status of any intracellular molecular pathway, 24 h after PEF-5 exposure, we subjected sham and pulsed cells to whole-transcriptome analysis. Interestingly, gene set enrichment analysis (GSEA) performed independently in each cell line showed a common upregulation of very few gene sets upon PEF-5 exposure (Figure 6a). Moreover, we could notice a partial overlap of downregulated processes only between NHA and D283 cells, with U87 NS cells not displaying any significant downregulated pathway (Figure 6b). This result suggested that each cell line could have engaged its specific transcriptional program in response to PEF-5. In particular, in NHAs, PEF-5 exposure significantly downregulated several processes involved in mitosis, chromosome maintenance, DNA replication and repair, and in extracellular matrix structure. Moreover, as a potential oxidative stress response, the *nuclear factor erythroid 2-related factor 2 (NRF2) pathway* and intracellular *ferroptosis*, in association with the *iron uptake and transport pathway*, were upregulated. In addition, the *nucleotide salvage pathway* was activated to recover bases and nucleosides from RNA and DNA degradation (Figure 6b). In D283 cell line, these transcriptional effects were further amplified, revealing after exposure a strong downregulation of *cell cycle* regulation and *DNA replication and repair*, and a significant reduction in *chromatin modification and epigenetic* processes. In contrast, there was a significant upregulation of *ferroptosis* and the *NRF2 pathway*.

As for U87 NS cells, GSEA did not evidence any downregulated pathway in pulsed cells. However, we could observe a significant general overactivation of RNA-related processes, including *tRNA processing* and *mRNA maturation and translation* (Figure 6c).

## 3. Discussion

PEF-based technologies are considered a promising clinical tool in cancer therapy [6]. The ability to cause harmful biological effects on tumors is evaluated by either directly inducing cytotoxicity through PEF exposure or indirectly through the introduction of molecules, therapeutic drugs, or genes into tumor tissues [36]. In this context, our preview studies showed that a specific pulse protocol (PEF-5: 0.3 MV/m, 40 μs, 5 pulses) was able to selectively target CSCs and activate different cellular processes, preserving NHAs [14,15]. In particular, in MB CSCs, PEF-5 exposure induced irreversible membrane permeabilization in association with a high level of apoptosis and senescence via the upregulation of GADD45A, inducing in vivo radiosensitization and resulting in the complete inhibition of tumor growth in combination with radiation exposure [14]. On the contrary, in GBM CSCs, PEF-5 exposure induced reversible electroporation, resulting in a low level of cell death and induction of neuronal differentiation associated with radioresistance, though reducing cell migration and clonogenic capacity [15]. These previous results suggested the possibility of developing ad hoc electrically mediated therapies for each type of tumor-targeted cell by modifying the specific parameters of PEF to neutralize CSCs and eradicate tumors while preserving normal cells in the surrounding tumor area. To reach this ambitious goal, it is essential to further investigate the mechanisms of the achieved selectivity on the various cell types. Because the predominant effect of PEF is the modification of the permeability of the cell membrane [36], an ultrastructural analysis in combination with a molecular/omics approach was adopted in this paper, to explain the vulnerability or resistance mechanisms triggered in different pulsed cells. 

Electron microscopic analyses confirmed that the main targets of PEF are the cell membrane and cytoskeleton, independently, on the exposed cells. For the first time, using a high-resolution electron microscopy technique, we observed membrane filopodium-like protrusion disappearance on the pulsed cell surface (Figure 1, Figure 2 and Figure 3), probably triggered by a strong ionic imbalance (calcium influx) induced by PEF-5 application or by the activation of actin depolymerization [5,37,38]. The current literature has never addressed PEF action on membrane filopodium-like protrusions. However, there is a general reference to cytoskeleton and actin modifications, such as fiber shortening, tightening, fragmentation, sparkling alteration, misalignment, and depolymerization [39], using different methodologies. A smoother and more homogeneous membrane was specifically reported [40] and, on the contrary, an increase in microvilli density was observed post exposure [41,42]. Other PEF-mediated effects described in the literature include cell swelling and blebbing [43]. Cell rounding with the presence of speckled actin spots and cell shape change are also mentioned [44,45]. It appears from the current literature that cell swelling can lead to cytoskeletal disruption. However, several studies also indicate cytoskeletal disruption in the absence of cell swelling, suggesting that additional mechanisms could be involved in this effect. Interestingly, in our investigation, we observed consistent cell swelling in NHAs and U87 NS cells (30% and 44%, respectively) and reduced cell swelling in D283 cells (11%) associated with ER enlargement proportional to cell swelling (Figure 1 and Appendix A). 

Some important differences were revealed in morphologic SEM and TEM analysis. Pulsed NHAs show a strong increase in MVB formation (a 14-fold increase, Appendix A), suggesting the activation of a survival mechanism [14,46]. This hypothesis is also supported by the upregulation of the *nucleotide salvage pathway* (Figure 6c), which permits the recovery of bases and nucleosides (recruited in intracellular vesicles) from RNA and DNA degradation. Pulsed NHAs could activate this strategy to repair the damaged area of the membrane and recover from the temporary disruption to maintain their vital functions [14,47,48]. Concomitantly, this notable rise in the occurrence of MVB formation observed in TEM images could be a potential response to oxidative stress, previously observed in NHAs by the authors [14]. Transcriptomic analysis showing the upregulation of *iron uptake and transport* (Figure 6c) suggests the activation of a protective mechanism to buffer iron excess [49,50,51]. The significant *NRF2 pathway* activation (Figure 6c) could contribute to regulating intracellular redox homeostasis in contraposition to *ferroptosis* (Figure 6c), iron-dependent non-apoptotic cell death [52]. Further, GSEA showed a significant downregulation of several processes involved in cell replication, such as mitosis, chromosome maintenance, DNA replication and repair, and extracellular matrix structure, confirming the cell cycle block and inhibition of proliferation that we previously described [14]. 

Conversely, in MB and GBM CSCs, TEM analysis highlights a different impact on the intracellular compartment, suggesting mitochondria are the elective target of PEF-5. Mitochondria alterations are more prominent in MB than GBM CSCs, in correlation with the higher percentage of CD133-positive cell proportion in D283 cells (NHA: 15%; D283: 90% [4]; U87 NS: 29%; Appendix A). Some studies indicate a functional role for CD133 in the reactive oxygen species (ROS) defense mechanism in evading anticancer therapies [53] and through functional cooperation with *NRF2* (Figure 6c) in sustaining quiescence and self-renewal abilities [54,55]. The significant reduction of CD133-positive cell after PEF-5 treatment (Figure 2), as a consequence of filopodia loss in D283 cells, suggests that it could alter tumor progression by making mitochondria more vulnerable to oxidative stress [53]. A PEF-5-mediated CD133 decrease could trigger severe damage to mitochondria, as observed in TEM images of D283 cells, leading to their complete dysfunction due to disruption in the electrochemical potential across the mitochondrial inner membrane and to a massive intracellular ATP depletion (90% reduction, Figure 5), resulting in mitochondrial cell death as a vulnerable response of MB CSCs. Bioinformatic analysis showed a significant downregulation of several processes involved in *cell cycle regulation* and *DNA replication and repair*, together with a significant reduction in *chromatin modification and epigenetic processes*, as a consequence of *ferroptosis* induction (Figure 6c), beyond the cell cycle arrest and the active apoptosis and senescence processes previously observed [14]. In these cells, the significant upregulation of the *NRF2 pathway* is probably insufficient to contrast *ferroptosis* activation (Figure 6c) [56,57]. In pulsed GBM CSCs (U87 NS), the levels of mitochondrial membrane depolarization and poor decrease in ATP content confirm that PEF-5 induces a mild perturbation in mitochondria (Figure 4 and Figure 5) that is insufficient to impair cell functions. In fact, transcriptomic results confirm that, at 24 h after exposure, GBM CSCs maintain their principal vital processes (including *tRNA processing*, *mRNA maturation*, and *translation*) (Figure 6c), suggesting that PEF-5 perturbation is transient and reversible. In this case, a lower level of CD133-positive cell and negligible alterations after PEF-5 exposure (Figure 2) may not be able to permanently alter the response to ROS [15], leading to massive cell death, as seen in D283 cells [14]. Altogether, these findings suggest that the impact of PEF-5 on mitochondria seems correlated with the content of CD133. The reduction/alteration of CD133 content could contribute to the induction of CSC mitochondrial death [21].

CSC mitochondria have been shown recently to be an important target for cancer treatment [58,59]. Mitochondrial function and energy metabolism play an important role in the acquisition and maintenance of CSC properties [55]. CD133 is involved in adaptive changes in cellular bioenergetic metabolism, providing CSCs with increased survival advantages [21]. PEF-5-induced cytoskeleton alterations (filopodium-like protrusion disappearance) and the reduction of CD133-positive cell lead to severe damage to mitochondria/ATP depletion (Figure 4 and Figure 5), consequently activating high levels of cell death in CD133-enriched cells such as D283 [14]. ATP depletion has been demonstrated to influence cytoskeletal dynamics [60,61]. This could serve as another contributing factor in the observed alterations in cytoskeletal response in our study (Figure 3 and Appendix A). Notably, ATP depletion is consistently significant in D283 cells (Figure 5). Hence, both cell swelling and ATP depletion play crucial roles in synergistically mediating cytoskeleton disruption in D283 [62]. In our cell models, the mitochondrial dysfunction and ATP depletion seem related to CD133-positive cell content, with a stronger effect on D283 cells. 

The molecular mechanisms underlying the cell response to PEFs and their outcome in terms of damage, repair, and cell survival or death are far from fully understood but the present work sheds some light on specific dysregulated signaling cascades proceeding from the cell surface to downstream effectors; CD133 action could be paradigmatic of other transmembrane proteins, perhaps cell-type or cancer-specific. Evidently, targeting mitochondria with PEFs, in correlation with CD133 content, may be crucial for developing potential therapeutic strategies. 

## 4. Materials and Methods

### 4.1. Cell Cultures

Human D283Med cell line (D283) was obtained from the American Type Culture Collection, (Manassas, VA, USA). Cells were routinely maintained in a complete growth medium (Eagle’s Minimum Essential Medium) supplemented with 10% fetal bovine serum (FBS), 2 mM glutamine, and 100 U penicillin/0.1 mg/mL streptomycin. The NHAs’ primary culture and optimized growth medium were purchased from Lonza (Basel, Switzerland). The human U87-MG cell line was purchased from the American Type Culture Collection (VA, USA) and was maintained in Dulbecco’s Modified Eagle’s Medium–high glucose (DMEM) containing 10% (*v*/*v*) FBS, 100 units/mL penicillin, and 100 mg/mL streptomycin. Cells were incubated at 37 °C with 95% air and 5% CO_2_. U87-derivated neurospheres (U87 NS) were cultured in a defined serum-free neural stem-cell medium (supplemented with 10 ng/mL of insulin, 2 mg/mL of heparin, 9.6 ng/mL of putrescine, 0.063 ng/mL of progesterone, 5 nM of HEPES, B27 1X without vitamin A, 20 ng/mL h-FGF, and 20 ng/mL of h-EGF; provided by Sigma Aldrich, Burlington, MA, USA) to enrich U87 cells in CSC content.

### 4.2. Cell Exposure to PEF-5

Cells were exposed to pulsed electric fields by electroporation cuvettes (0.1 cm gap, Bio-Rad Laboratories Srl, Milan, Italy) connected to a pulse generator (Schaffner NSG504, Luterbach, Switzerland), as fully detailed in our previous work [14]. Electric pulses of exponential shape were used in our experiments, and their amplitude and duration were maintained at 300 V (for an electric field of 0.3 MV/m) and 40 μs at full width at half maximum, respectively. Cells were exposed to 5 pulses, and an interpulse interval was equal to 1 Hz (PEF-5). Monitoring of electric pulse voltages was carried out as fully described in a previous study [14].

### 4.3. Electron Microscopy

#### 4.3.1. SEM

For scanning electron microscopy (SEM), samples were fixed for 4 h at 4 °C with 2.5% (*v*/*v*) glutaraldehyde + 2% (*v*/*v*) paraformaldehyde in 0.1 M cacodylate buffer, pH 7.2. After 3 × 20 min washings at 4 °C in the same buffer, samples were post-fixed with 1% (*v*/*v*) osmium tetroxide (Agar Scientific Ltd., Stansted, UK) in 0,1 M cacodylate buffer, pH 7.2, for 2 h at 4 °C. Specimens were then washed in the same buffer (3 changes for 15 min each at 4 °C) and dehydrated in a graded ethanol series. Samples were dried by the critical point method using CO_2_ in a Balzers Union CPD 020 unit (Balzers Union Limited, Balzers, Liechtenstein). Then, samples were attached to aluminum stubs using carbon tape and sputter-coated with gold in a Balzers MED 010 unit (Balzers, Liechtenstein). The observations were made using a JEOL JSM 6010LA electron microscope (JEOL Ltd., Kranj, Slovenia).

#### 4.3.2. TEM

For ultrastructural analyses by TEM, samples were fixed and dehydrated, as described above. After 3 changes in 100% ethanol (10 min each), 2 changes in propylene oxide followed for 10 min each at 4 °C. Samples were then infiltrated with mixtures of Agar 100 resin/propylene oxide in different percentages. At the end of the procedure, samples were embedded in pure Agar 100 resin and left to polymerize for 2 days at 60 °C. Resin blocks were cut with a Reichert Ultracut ultramicrotome using a diamond knife. 

#### 4.3.3. Pre-Embedding Labeling of Cell-Surface Antigens for Electron Microscopy

Cells were prefixed with 4% (*v*/*v*) paraformaldehyde + 0.1% (*v*/*v*) glutaraldehyde in PBS, pH 7.4, at RT, for 10 min. Samples were then washed in PBS (pH 7.4) containing 50 mM glycine to quench aldehydes. A block step was made for 30 min in 1% bovine serum albumin (BSA) and 10% normal goat serum (NGS) (British Biocell International, Cardiff, UK) in PBS (pH 7.4), then cells were washed in 1% BSA in PBS. Cells were incubated with the primary antibody, CD133 (AC133, Miltenyi Biotec, Bergisch Gladbach, DE, Germany) diluted 1:10 in 1% NGS, 0.1% Tween 20, 1% BSA, and 0.1% sodium azide in PBS pH 8.2 (buffer A) for 3 h. Cells were then washed twice in buffer A and incubated for 1 h in a moist chamber with a secondary gold-labeled goat anti-mouse antibody (gold particles of 10 nm diameter) (British Biocell International) diluted 1:10 in buffer A. After two rinses in PBS, cells were fixed in 1% glutaraldehyde in PBS for 20 min at 4 °C and washed twice in distilled H_2_O. Samples were then post-fixed in 0.5% osmium tetroxide in distilled H_2_O for 30 min and washed twice in distilled H_2_O. Cells were dehydrated with a graded ethanol series and then infiltrated with mixtures of LRWhite resin/ethanol in different percentages. At the end of the procedure, samples were embedded in pure LRWhite resin (Agar Scientific) and left to polymerize for 2 days at 50 °C. Tightly capped gelatine capsules were used to allow resin polymerization. Ultrathin sections were obtained, as previously described. For TEM analysis, ultrathin sections (60–80 nm) were collected on copper grids, stained with uranyl acetate and lead citrate, and observed with a JEOL 1200 EXII electron microscope (JEOL Ltd., Kranj, Slovenia). Micrographs were captured by an Olympus SIS VELETA CCD camera (Münster, Germany) equipped with iTEM 5.1 software.

### 4.4. F-Actin Staining Protocol

Alexa Fluor 488 phalloidin (Thermo Fisher Scientific, Rome, Italy) was used to visualize F-actin in NHAs, D283 cells, and U87 NS cells. Cells were fixed with 4% (*v*/*v*) paraformaldehyde, 24 h after PEF-5 exposure. After incubation with 0.1% Triton X-100 in PBS, cells were incubated for 30 min in 1% BSA to reduce nonspecific background. The phallotoxin staining solution was incubated for 20 min at RT. After washing in PBS, Vectashield Coverslip Mounting Solution with DAPI (Vector Laboratories, Inc. Burlingame, CA 94010, USA) was dispensed on samples for nuclear counterstaining and preserving fluorescence. Cell images were acquired using an Oxio Observer Inverted microscope (Zeiss, Oberkochen, Germany) equipped with an Apotome 3 imaging system (Oberkochen, Germany).

### 4.5. Morphometric Analyses

Morphometric analyses were evaluated using Olympus iTEM software. In particular, to compute the percentage of damaged mitochondria (with disorganized stroma and cristae) per cell, 50 cells/sample were evaluated. In addition, a transversal area of 10 mitochondria per cell was measured in 10 cells per sample group. To evaluate ER width, 10 measurements per cell in 10 samples per group were taken into account. The cell and vesicle area measurements were obtained from 50 cells per sample group. Results obtained by measurements of mitochondrial area, cell area, vesicle area, and ER width in pulsed cells were shown in graphs as fold change with respect to sham-exposed cells.

### 4.6. Transcriptional Analysis

NHAs, D283 cells, and U87 NS cells were subjected to whole-transcriptome analysis through Clariom S Affymetrix chips, 24 h after PEF-5 exposure (single-gene fold change is shown in Appendix A). In particular, in vitro transcription, hybridization, and biotin labeling of RNA were performed according to the GeneChip™ WT Kit protocol and Clariom™ S human gene platform (Thermofisher, Waltham, MA, USA). Microarray data (CEL files) were generated using default Affymetrix microarray analysis parameters (Command Console Suite Software v. 4.0+ by Affymetrix, Waltham, MA, USA) and normalized (independently for each cell line) through Transcriptome Analysis Console (TAC) software v. 4.0.2.15 (Waltham, MA, USA). Newly generated NHA and D283 expression data were deposited into the Gene Expression Omnibus (GEO) database under Series Accession Number GSE248601 and are accessible without restrictions. U87 NS expression data were obtained from U87 neurospheres included in our previously published series, GSE195506 [15]. Differentially expressed genes in PEF-5 exposed relative to sham samples (n = 4 for each group; Appendix A) were identified for each individual cell model by using Limma [63]. Accordingly, in order to identify the potential pathways and intracellular signaling affected by pulse exposure, we performed GSEA looking for differentially enriched (FDR < 0.05) pathways included in the MSigDB-C2cp gene set collection. Enrichment maps of the significantly enriched C2cp GSEA terms (FDR ≤ 0.05) in PEF-5 exposed cells were generated using the Enrichment Map application in Cytoscape 3.10.0.

### 4.7. Intracellular ATP Measurement

The CellTiter-Glo^®^ Luminescent Cell Viability Assay (Promega, Milan, Italy) was used to quantify the presence of intracellular ATP, which signals the presence of metabolically active cells. Luminescence was monitored at 30 min after PEF-5 exposure using a GloMax plate reader (Promega, Milan, Italy), according to the manufacturer’s instructions. Reactions were performed in triplicate for each biological replicate. Intracellular ATP was calculated as luminescence fold change with respect to the relative control for each cell culture.

### 4.8. Mitochondrial Membrane Potential Evaluation

A MitoProbe™ JC-1 Assay Kit (Thermo Fisher Scientific, Waltham, MA, USA) was used to highlight mitochondrial membrane potential disruption. The protocol describes the introduction of JC-1 reagent into cultured cells and the analysis of the stained cells by flow cytometry, according to the manufacturer’s instructions. JC-1 exhibits potential-dependent accumulation in mitochondria, indicated by a fluorescence emission shift from green (~529 nm) to red (~590 nm). Consequently, mitochondrial depolarization is indicated by a decrease in the red/green fluorescence intensity ratio. The potential-sensitive color shift is due to the concentration-dependent formation of red fluorescent J-aggregates. Data were processed using FCS Express 7 Flow. 50,000 events were acquired. FL2 (585/30)-FL1 (525/40) compensation was set to 40%. The data reported are the mean of results obtained from three independent experiments.

### 4.9. CD133 Quantification

The percentage of CD133-positive cell was evaluated in NHA, D283, and U87 NS cells by cytometric analysis. One hour and 24 h after PEF-5 exposure, D283 and U87 NS cells were stained with PE-conjugated mouse anti-CD133 antibody (2 μL/10^6^ cells; AC133, Miltenyi Biotec, Bergisch Gladbach, DE or sc-365537, Santa Cruz Biotechnology, Dallas, TX, USA) for 10 min. Samples were analyzed using a CytoFLEX flow cytometer (Beckman Coulter, Brea, CA, USA). Data are presented as percentages of positive cell in the live-gated cell population, as determined by physical parameters.

### 4.10. Statistical Analysis

Results are expressed as a mean of three biological replicates ± SEM. All statistical tests were performed with GraphPad Prism software v.7 (GraphPad, CA, USA). *p* values were determined using a two-tailed *t* test; * *p* < 0.05; ** *p* < 0.01; *** *p* < 0.001; **** *p* < 0.0001. 

## 5. Conclusions

Ultrastructural analysis, supported by molecular/omics characterization, was performed to gain a better understanding of the selectivity of PEF-5 exposure on brain CSCs and NHAs. Overall, PEF-5 exposure induces cell swelling, leading to cytoskeleton alterations (filopodium-like protrusion disappearance) and, where a high CD133 content is present, we observed its decrease due to protein localization on the lost membrane protrusions. This decrease contributes to mitochondrial dysfunction and ATP depletion in D283 cells, which further confer cytoskeleton alterations. This study could serve as a starting point for targeted anticancer therapy in highly CD133-positive tumors. Further studies are undoubtedly necessary to validate our hypothesis, especially in the context of CSCs derived from other tumor types, in order to confirm its universality and potential implications for future therapies.

## Figures and Tables

**Figure 1 ijms-25-02233-f001:**
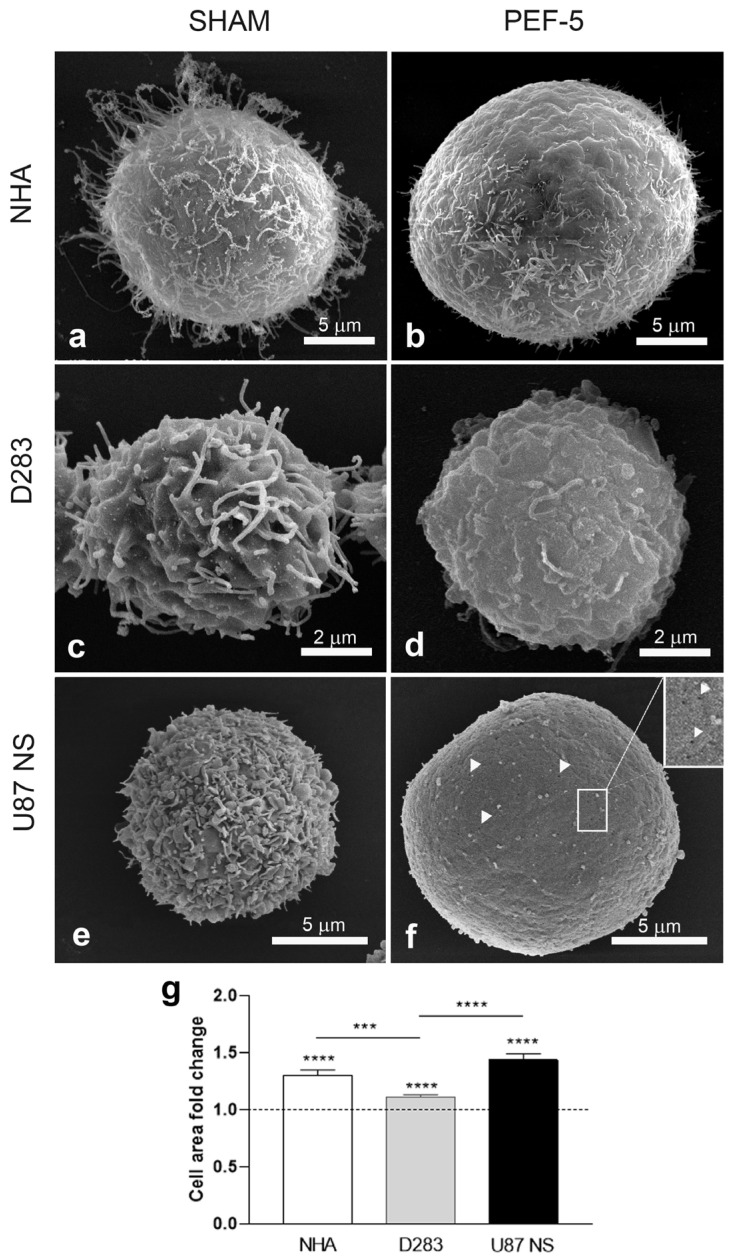
Ultrastructural analysis by SEM. Sham or PEF-5-exposed SEM images of (**a**,**b**) NHA, (**c**,**d**) D283, and (**e**,**f**) U87 NS cells. Filopodia protrusions disappeared in all analyzed cells 1 h after PEF-5 exposure, revealing visible pores on the surface of U87 NS cells, as indicated by the white arrowheads (**f**) and relative magnification (inset). (**g**) The graph shows the cell area fold increase of PEF-5-exposed cells with respect to relative sham-exposed cells (dotted line). *p* values were determined using a two-tailed *t* test; *** *p* < 0.001; **** *p* < 0.0001.

**Figure 2 ijms-25-02233-f002:**
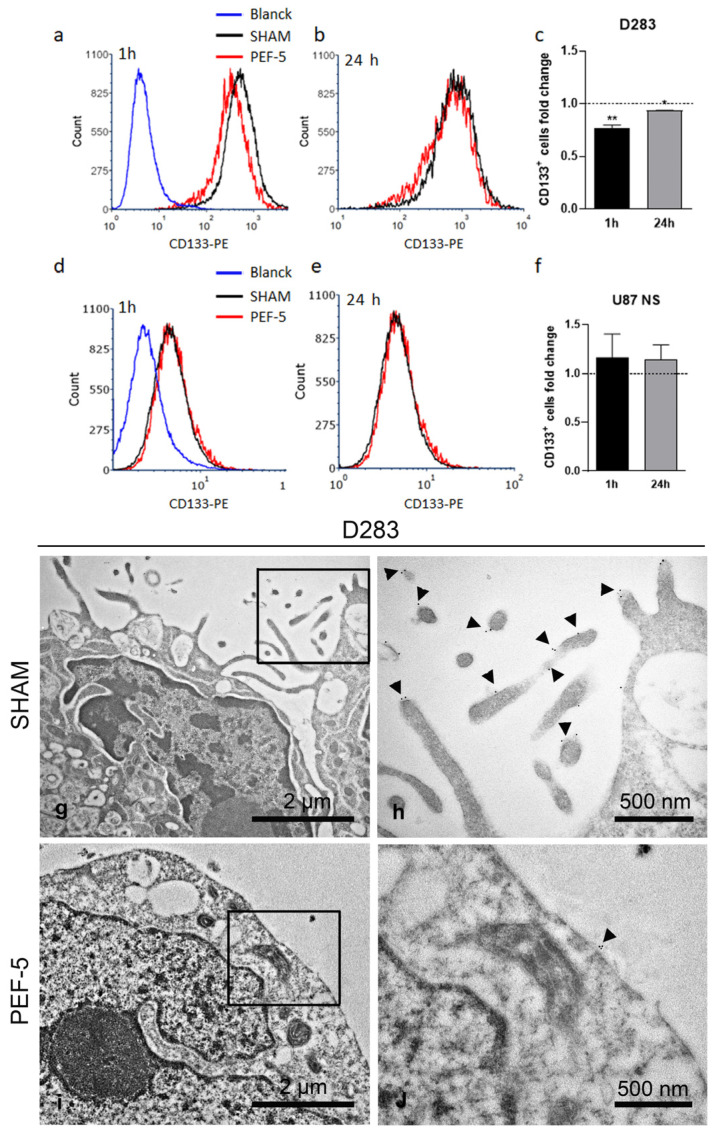
Cytofluorimetric quantification of CD133-positive cell and immunogold electron microscopy (IEM) analysis. Representative histograms of anti-CD133 staining and relative quantification of (**a**–**c**) sham- and PEF-5-exposed D283 cells and (**d**–**f**) U87 NS cells. IEM analysis in (**g**,**h**) sham-exposed D283 cells shows the normal localization of CD133 protein on plasma membrane protrusions, as indicated by the black arrowheads; (**i**,**j**) no filopodium-like protrusions and rare gold particles along the flat regions of the plasma membrane were observed in pulsed D283 cells 1 h after PEF-5 exposure. *p* values were determined using a two-tailed *t* test; * *p* < 0.05; ** *p* < 0.01.

**Figure 3 ijms-25-02233-f003:**
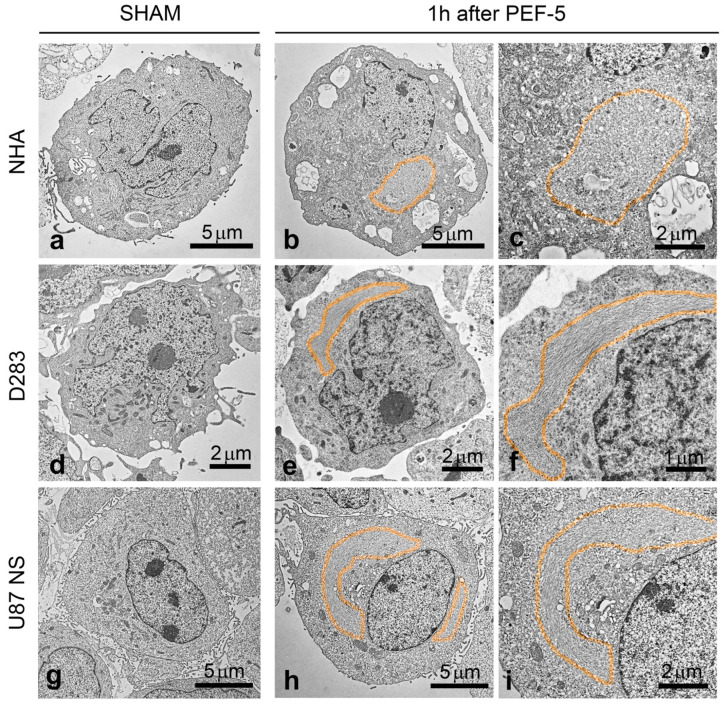
TEM images of cytoskeleton distribution 1 h after PEF-5 exposure. TEM images show sham and pulsed (**a**–**c**) NHA, (**d**–**f**) D283, and (**g**–**i**) U87 NS cells. The colored area highlights the region where the cytoskeleton is distributed and densely packed, and in (**c**,**f**,**i**), the relative magnification shows fine details.

**Figure 4 ijms-25-02233-f004:**
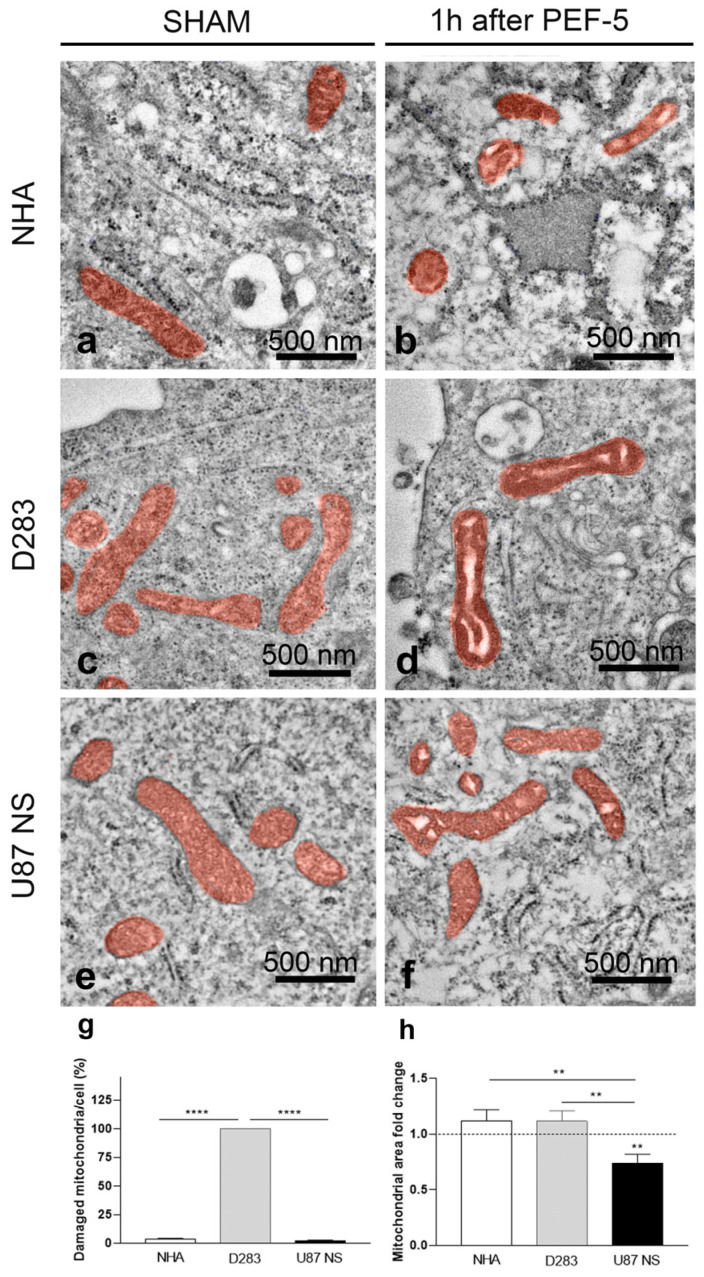
Mitochondrial morphological alteration assessment 1 h after PEF-5 exposure. (**a**–**f**) In TEM images, mitochondria are colored in red to highlight the difference between sham- and PEF-5-exposed (**a**,**b**) NHA, (**c**,**d**) D283, and (**e**,**f**) U87 NS cells. (**g**) The graph shows the percentage of morphologically damaged mitochondria per pulsed cell. (**h**) The graph shows the transversal mitochondrial area in pulsed cells compared with that of relative control cells (dotted line). *p* values were determined using a two-tailed *t* test; ** *p* < 0.01; **** *p* < 0.0001.

**Figure 5 ijms-25-02233-f005:**
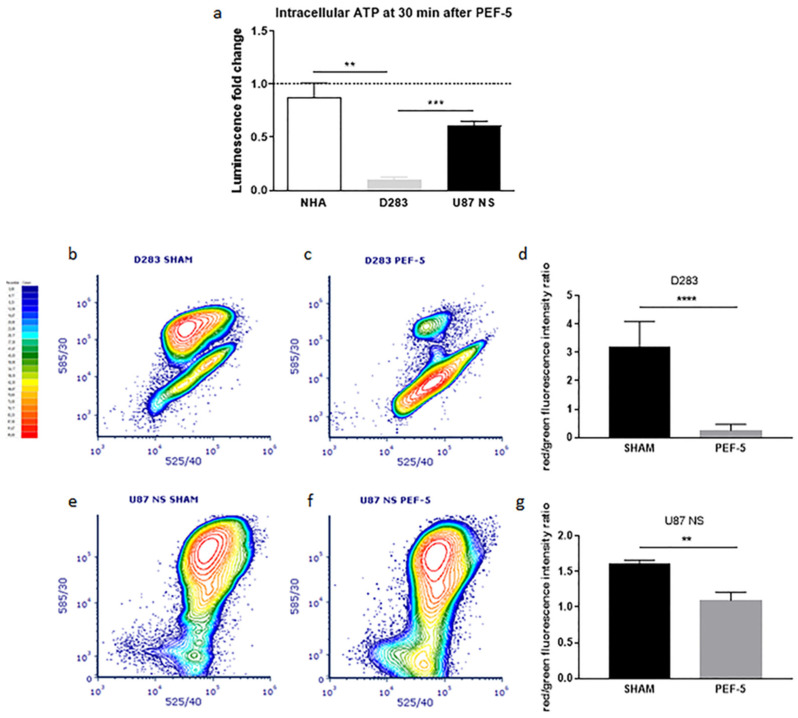
Mitochondrial dysfunction assessment. (**a**) The graph shows the intracellular ATP quantification in NHA, D283, and U87 NS cells 30 min after PEF-5 exposure. Intracellular ATP was calculated as luminescence fold change with respect to relative control for each cell type. (**b**–**g**) Cytometric analysis of JC-1 in untreated cells and after PEF-5 exposure. 50,000 events were acquired. FL2 (585/30)- FL1 (525/40) compensation was set to 40%. Densitometric analyses (**b**,**c**,**e**,**f**) are representative examples of biological triplicates. The color scale indicates the relative percentile. (**d**,**g**) Graphs show the mean red/green fluorescence intensity ratio ± SD in D283 and in U87 NS, respectively. *p* values were determined using a two-tailed *t* test; ** *p* < 0.01; *** *p* < 0.001; **** *p* < 0.0001.

**Figure 6 ijms-25-02233-f006:**
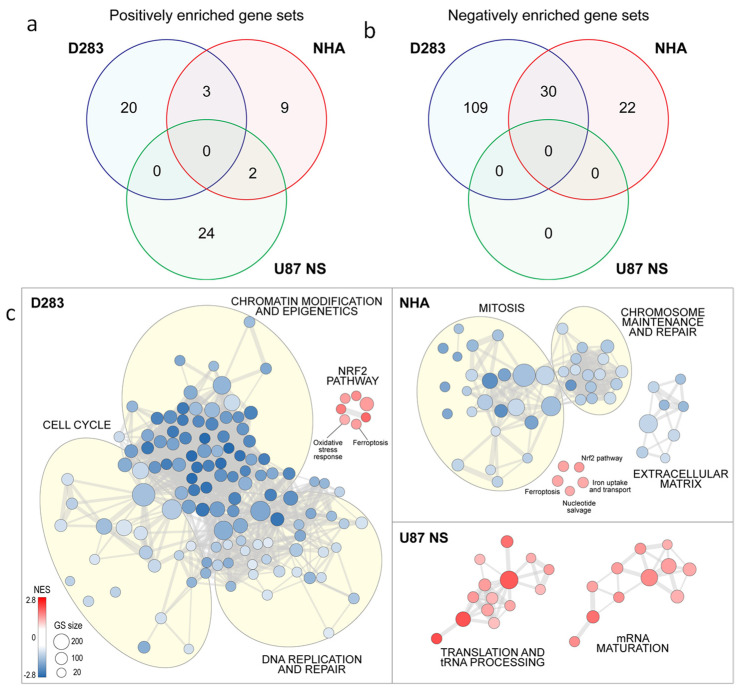
Transcriptomic analysis, evaluated 24 h after PEF-5 exposure. (**a**,**b**) Venn diagrams show commonly enriched pathways from C5bp (biological process) gene sets. (**c**) Enrichment map displays the significantly (FDR q value < 0.05) enriched pathways (up- and downregulated) in PEF-5 treated NHA, D283, and U87 NS cells, relative to matched controls (C2cp MSigDB). GS: gene set; NES: Normalized Enrichment Score.

## Data Availability

All data are listed in tables or presented in figures in the main text or Appendix A. Moreover, gene expression data generated within this study have been deposited into the GEO database under Series Accession Number GSE248601and are accessible without restrictions.

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
