# Peer review of "Involvement of Mitochondria in the Selective Response to Microsecond Pulsed Electric Fields on Healthy and Cancer Stem Cells in the Brain"

_ijms, 2024, doi:10.3390/ijms25042233_

Round 1
Reviewer 1 Report
Comments and Suggestions for Authors
In this study, Tanori and colleagues intended to decipher cellular responses of brain cancer stem cell (CSC) lines to 40 microsecond pulsed electric field at 0.3 MV/m. They compared the pulse effects among two CSC lines and normal astrocytes. They performed various analyses, including SEM, TEM, mitochondrial assays and transcriptome analysis. Overall, the authors have made significant efforts, but many concerns can be found in the manuscript. In Results, there are logical leaps that inadequately connect experimental observations with their interpretations. Some of TEM images are unclear and too small to draw conclusions. Quantitative data on organelle/subcellular structures are required to validate their interpretations of TEM images. Discussion is composed of excessive speculation but lacks important arguments on their own results. The title is vague and should be in a more focused way. Abstract does not adequately convey the content of this study. Specific points are as follows:
1. Introduction
lines 89-91 "For these reasons, the targeted mitochondrial functionality by PEF-5 in addition to explain the selectivity action of the electric exposure on CSCs and NHA could be also exploited as a powerful strategy to elicit CSCs death."
The meaning of this sentence is not clear.
2. Results
line 99 : filopodia
(1) A brief explanation for filopodia in this paragraph or Introduction would be helpful for the readers.
(2) The shapes of membrane protrusions greatly differ among three cell lines (Fig 1 a, c, d). Is it biologically appropriate to regard these three types with different morphology as filopodia? If so, why?
(3) If any previous studies have reported the effects of electric pulse on filopodia, they should be cited. If not, the authors can say that it is the first indication in this research field.
line 105 - 106 "Surprisingly, SEM analysis highlighted a similar impact on cell surface, despite the previously reported different molecular processes activated after PEF-5 exposure."
The meaning of this sentence is not clear.
Fig 1 f
Black arrowheads are difficult to see due to low contrast with gray background.
line 120-123
The authors need to clearly distinguish between expression level (= protein amount / cell) of CD133 and CD133-positive cell proportion to avoid confusion by the readers.
line 120 : "90% and 29%" in Sup Fig seem to be CD133-positive cells (proportion of cells), but not expression level (protein amount/cell). If it is true, this sentence may be changed to "CD133 was detected in 90% and 29% of D283 and U87 NS cells by flowcytometry"
line 121, 122 : "a significant decrease" (line 121) and "no significant perturbation" (line 122) seem to indicate protein amounts as estimated by flowcytometry. If it is true, this sentence needs more clarity.
Fig 2 f
The error bar is unnaturally small. Is it true?
Fig 2 d, e
If the horizontal axes indicate the fluorescence intensities of CD133, the expression of CD133 in U87 NS is more than two-order lower (more than 100-fold low) than that in D283 (Compare panel a and d, panel b and e). If so, the relevance of this experiment is seriously questionable, because U87 NS should be regarded to be cells without CD133 expression.
Based on the published study, the authors claimed that CD133 is not expressed in NHA (line 119). The authors need to perform the same analysis using NHA and compare CD133 fluorescence between NHA and U87 NS. By this, they can demonstrate that very weak fluorescence of CD133 in U87 NS is not background, and they can conclude that the pulse effect on CD133 in U87 NS is biologically meaningful.
Fig 2 g h
It is hard to rationalize that U87 NS was not included in the analysis. The authors should demonstrate that CD133 distribution is not affected by pulse treatment, which will validate the results of Fig 2 a b d e.
line 139
The title of this section is inadequate: "target" generally implies that electric pulse directly affects the mitochondria. The authors did not examine whether pulse effect is direct or secondary.
In addition, the authors did not validate the membrane structures are really exosomes. Furthermore, they just observed the membrane alterations in TEM images but did not examine neither trafficking nor activation.
line 146 "in NHA cells the ER appeared to be enlarged"
It is very hard to judge the size of ER from the images in this manuscript. Need quantification of ER size in comparison with untreated samples. Addition of more enlarged and highlighted images in Supplementary Info is very helpful for the readers.
line 147 "the mitochondria showed ultrastructural changes" (Figure 3c and f).
It is very hard to judge what are morphological changes in mitochondria from this panel. Again, addition of more enlarged and highlighted images in Supplementary Info is very helpful for the readers. Size quantification also needed.
lines 150-151 "a significant increase in multivesicular bodies (MVBs)/exosomes trafficking"
It is very hard to judge whether the observed structures are really exosomes or just mechanical damages that are frequently observed in pulsed cell membranes.
Even if they are exosomes, the authors did not examine their "trafficking" in this study.
lines 155-157
The authors repeatedly used qualitative words as follows:
line 155 rather less dense, apparently normal
line 156 generally smaller
line 157 only sporadically
The authors should perform the quantification of organelle sizes in TEM images and need to provide more convincing data, if they want to use the above expression.
Overall, the descriptions in the section 2.3 are too speculative without sufficient validity.
Fig 4
(1) Need data on untreated cells. Interpretations should be made in comparison with untreated cells.
(2) It is very hard to judge whether fibrous structures in the TEM images are really cytoskeleton or just artifacts due to heavy fixation for TEM. Immunofluorescence of cytoskeleton components may be more informative than TEM.
line 177
2.4. Not 2.3.
line 179 "milder in NHA and in 179 U87 NS"
At least, from the images in this manuscript, it is hard to draw this conclusion. Need quantitative data and enlarged/highlighted images as mentioned above.
line 205
2.5. Not 2.4.
line 208
When first appears, an abbreviation should be spell-out, although it is shown in line 418.
lines 209 - 213
Too long. Separate this long sentence into several ones
lines 216-223
many words are italicized. What do they means?
Fig 6
This figure does not provide sufficient information for the readers. The names of genes in each panel should be listed as tables in supplementary information. Fold changes of expression of individual genes should be included in the tables. The legend of Fig 6 missed panel c?
3. Discussion
line 239
context. Not contest.
lines 256-283, 284-318, 319-335
In these three paragraphs, the authors greatly stretched their imagination, although they presented limited sets of experimental data in this manuscript. Because excessive speculation should be strictly avoided, most of these paragraphs appear to be deleted. The authors should keep sentences concise in a scientifically logical way based on their experimental data in this manuscript.
I suggest that they should focus on their experimental results as follows:
1. filopodia
- How filopodium structure is maintained in a normal cell (with refs).
- A possible mechanism why pulse treatment causes the disappearance of filopodia. Discussion in relation with the membrane damage by electric pulse.
- Are there any previous studies that have reported the disappearance of filopodia by pulse treatment? If yes, the authors should cite them. If no, they can highlight the novelty of their findings in Fig 1 in the Abstract, Results, and Conclusion sections.
2. CD133
A possible mechanism that CD133 rapidly disappears from the membrane surface at 1 h after pulse treatment. Pulse may induce degradation of CD133 or membrane internalization. The authors can mention the recovery of the surface CD133 at 24 hr as compared with 1 hr (Fig 2c).
3. A possible mechanistic link between CD133 and mitochondria in pulsed cells
How CD133 on the cell surface is functionally connected to intracellular mitochondria.
The authors should cite appropriate literatures on this points. The current form of the author's discussion is not clear and does not explain sufficiently to convince the readers.
line 243
If the authors observed GADD45 upregulation in their previous study (ref 13), they should indicate where is GADD45 in Fig. 6, by which the authors can show the consistency between their previous and current studies.
line 263 "enlargement of the intraluminal ER" "a strong increase"
Inadequate expression. As written above, quantitative validation and presentation of more highlighted images are required if the authors want to say so.
"multivesicular bodies biogenesis"
Inadequate expression. The authors did not confirm whether biogenesis or just mechanical damage. Discussion on the relationship with autophagy is too much.
line 266
Why italicized?
4. Methods
lines 346, 351
FBS: Spell out first, abbreviation next.
line 406
What is "at point 2.2"
line 415
The accession number is strange.
line 416, 417, 420
Delete dashes
5. Conclusion
Their conclusions deviate the proper logical flow of science.
line 458 "underscores the role of mitochondria in cell response"
This is not true. The expression "the role of mitochondria in cell response" generally implies that the effects on mitochondria occur first, and this mitochondrial alterations cause other cell response.
The authors observed some changes in mitochondria but did not prove the direct relationship between mitochondrial dysfunction and other cellular responses.
line 459 "This role appears to be mediated by CD133, as its downregulation leads to the deregulation of ROS generation"
If the authors want to say something about ROS in Conclusion, they should perform experiments on ROS, such as ROS measurement, which is not a big experiment.
line 462 "we cannot rule out the possibility of a direct impact on mitochondrial membranes by PEF-5"
This sentence is inconsistent with the previous paragraph (line 458)
Title
The title should be more specific. "Insights into" is obscure and not informative.
In the main text, the difference between D283 and U87 NS is interesting and appears to be scientifically important. However, this title says the comparison between normal and cancer cells. So, the title does not reflect this study.
Abstract
Abstract, particularly the later half of it, should be thoroughly edited.
line 26 "presenting indiscriminate filopodia disappearance"
This sentence is hard to understand. More clear and plain explanation can be possible.
line 29 "NHA ..... eliminated the detrimental ROS"
In this study, the authors did not conduct any experiments on ROS, and thus, it is inadequate to mention ROS in Abstract. The authors should focus more on their achievement in this manuscript.
Comments on the Quality of English LanguageThe quality of English in this manuscript is OK, although there are several points to be corrected.
Reviewer 2 Report
Comments and Suggestions for Authors
Manuscript Review - Manuscript ID: ijms-2752375
“Insights into selective action of microsecond pulsed electric fields on healthy and cancer stem cells”
This manuscript investigates the impact of PEF-5 on cancer stem cells to observe subcellular structural changes, aiming to comprehend the vulnerability or resistance mechanisms triggered by PEF-5. A notable finding was the indiscriminate disappearance of filopodia on the cell surface, accompanied by mitochondrial damage responses associated with this event. The results are mainly supported by EM images and flow cytometry data. The reviewer offers the following comments and questions for improving the manuscript.
Comments
1. While Figure 2g includes multiple cells, Figure 2h only shows a part of one cell. The review suggests having a better representative image for 2h. Also, Figure 2 would be more comprehensive if it includes IEM data for U87 NS.
2. Figure 3 should use colored arrows and dashed lines to indicate mitochondria and swollen ER, respectively.
3. It would be more helpful for the reader to comprehend the mitochondrial outcome by having additional drawing or animation describing the difference between intact and disorganized mitochondria.
4. The change in cytoskeleton observed in TEM images (Figure 4) indicates that cytoskeleton is a target of PEF treatment. The reviewer suggests having additional data showing cytoskeleton changes at the mRNA and/or protein levels.
5. Aligning with the in-dept study of mitochondrial dysfunction upon PEF treatment as shown in Figure 5, the study should have an additional in-dept functional study of the ER as well.
6. Figure 6 legend should be revised. (a) should be (a-b), and (b) should be (c).
7. In the discussion, the basal level of CD133 in U87 NS lacks reference (line 289). Also, in the method, it seems that U87 NS is induced to enrich CSC contents. The reviewer is interested to see supplement quantification data of CD133 in U87NS and whether it is consistent batch to batch.
8. It is also not clear whether the three cell models chosen for this study since they have different levels of CSC: NHA (none), MB D283 (high), and U87NS (low), and whether CD133 expression directly reflects the CSC ratio in each cell model. If this is an important rationale of using these cell models and the levels of CD133, the authors should clarify this in the introduction more explicitly.
Questions
1. Why did the author choose to measure ATP levels after 30 minutes treated with PEF (Figure 5) while other experiments (Figure 1, 2, 3, 4) examined the effect of PEF after 1 hour treatment?
2. The activation and downregulation of different cellular processes were accessed simply via bioinformatics data. Would the authors consider having additional evidence from experimental observations to strengthen the findings?
Round 2
Reviewer 1 Report
Comments and Suggestions for Authors
The manuscript was sufficiently improved.